# Biomedical Signals for Healthcare Using Hadoop Infrastructure with Artificial Intelligence and Fuzzy Logic Interpretation

Shitharth Selvarajan [1,*], Hariprasath Manoharan [2,*], Tawfiq Hasanin [3], Raed Alsini [3], Mueen Uddin [4], Mohammad Shorfuzzaman [5,*] and Abdulmajeed Alsufyani [5]

1 Department of Computer Science & Engineering, Kebri Dehar University, Kebri Dehar 001, Ethiopia
2 Department of Electronics and Communication Engineering, Panimalar Institute of Technology, Poonamallee, Chennai 600123, India
3 Department of Information Systems, Faculty of Computing and Information Technology, King Abdulaziz University, Jeddah 22254, Saudi Arabia; thasanin@kau.edu.sa (T.H.); ralsinie@kau.edu.sa (R.A.)
4 School of Digital Science, University Brunei Darussalam, Jalan Tungku Link, Gadong BE1410, Brunei; mueenmalik9516@gmail.com
5 Department of Computer Science, College of Computers and Information Technology, Taif University, Taif 21944, Saudi Arabia; a.s.alsufyani@tu.edu.sa
* Correspondence: shitharth.it@gmail.com (S.S.); hari13prasath@gmail.com (H.M.); m.shorf@tu.edu.sa (M.S.)

**Abstract:** In all developing countries, the application of biomedical signals has been growing, and there is a potential interest to apply it to healthcare management systems. However, with the existing infrastructure, the system will not provide high-end support for the transfer of signals by using a communication medium, as biomedical signals need to be classified at appropriate stages. Therefore, this article addresses the issues of physical infrastructure, using Hadoop-based systems where a four-layer model is created. The four-layer model is integrated with Fuzzy Interface System Algorithm (FISA) with low robustness, and data transfers in these layers are carried out with reference health data that are collected at various treatment centers. The performance of this new flanged system model aims to minimize the loss functionalities that are present in biomedical signals, and an activation function is introduced at the middle stages. The effectiveness of the proposed model is simulated by using MATLAB, using a biomedical signal processing toolbox, where the performance of FISA proves to be better in terms of signal strength, distance, and cost. As a comparative outcome, the proposed method overlooks the conventional methods for an average percentage of 78% in real-time conditions.

**Keywords:** biomedical signals; Hadoop systems; healthcare; fuzzy interface system; optimization

## 1. Introduction

The application of biomedical signals in several distributed systems has gained importance in recent years, and in order to observe its major characteristics, it is applied in medical data analysis. Since the early detection of various diseases is necessary, it is essential to apply biomedical signals where rapid response can be achieved. The same process can be carried out by using any wireless modules, but biomedical signals are applied in a direct form to monitor the conditions of the body, so it is better to implement it in all distribution systems, using a Hadoop infrastructure. The major advantage of the Hadoop infrastructure is that, since distribution systems are used, the data will be distributed in an automated way to other users. In this regard, security of data processing with all parallel mechanisms will result in failure; this achieves much low efficiency in medical applications. Thus, to prevent a high failure rate, the distribution systems that process the data by using a Hadoop infrastructure are tested by using an open-source data framework. Therefore, the entire system prevents mismanagement of data without any reproduction at output states, and the data framework at the receiver will process in an even performance fashion.

Figure 1 demonstrates the process of the proposed method by using different gestures from patients, and it is preprocessed by avoiding all complex systems. The gestures are preprocessed by using only the supported system, as it is applied at pattern-recognition schemes for further processing. In this stage, biomedical signals are processed with the integration of a four-layer Hadoop infrastructure, which is further controlled by using localization units. Once the units are controlled, then a data representation framework with master and slave blocks is represented for managing the input resources with maximization of signal distance using corresponding nodes. Then, at the final stage, the nodes are managed and applied with high-security features for attaining output data.

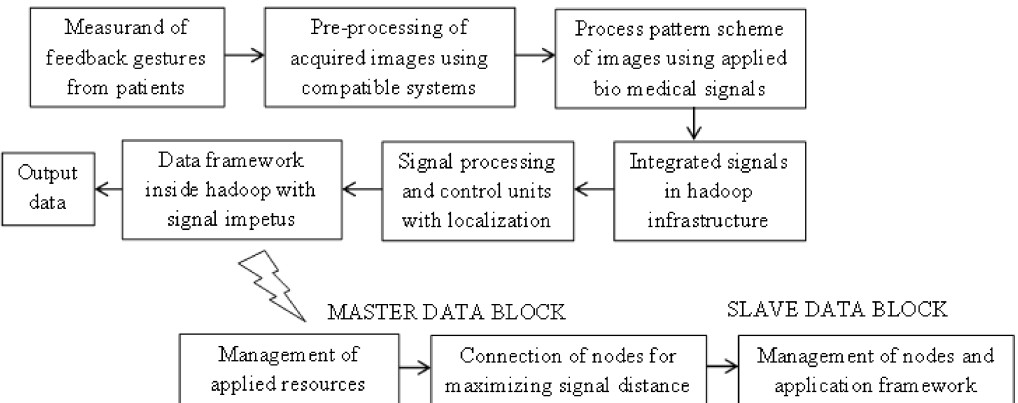

**Figure 1.** Block diagram of signal representation and processing using Hadoop infrastructure.

### 1.1. Literature Analysis

To have a basic knowledge of the existing Hadoop system, many conventional supporting models are analyzed in this section. Since many systems are having their own drawbacks, multiple solutions are given for attainment progress by some other researchers. Therefore, an overview of the system structure for biomedical signals with Hadoop and fuzzy logic system is examined. In Reference [1], a classification mechanism was identified where diseases were diagnosed with many sensing applications. Even big data are integrated in the classification mechanism for observing effective results under four different layering structures. However, in the visual representation, the recovery of data paths is not possible; thus, a correlation exists between different variable ciphers, and to store data segments, a computing process is not formed. To store the data segments in a clear format, a cloud computing system was created [2], and fuzzy set rules were applied with membership functions. In addition, the classified data segments are demonstrated by using a decision table mechanism, using distinct classification time. However, with cloud computing, more analytical rules are created with high false rates, thus, in turn, reducing the computational speed of Hadoop systems. Furthermore, a systematic review was accompanied by many existing models that support the fuzzy system, thus creating a centralized system for patients inside the application areas [3].

It is considerable that big data tools are needed to a large extent to support multiple systems that exist in biomedical signal process, thus boosting the strength of signals without any interruption. In addition, the procedure of Hadoop is examined in a social networking database to visualize the impact of streaming which can be converted to biomedical signal representation cases [4]. However, some of the specified factors will not extend great support, as high domain knowledge is needed in learning phases. The data storage using Apache provides an additional advantage to Hadoop systems, but the supporting cost of the system will be increased to a great extent. Hence, a complete examination was carried out with big data analytics, using several forms and structures [5]. If any system integration is needed, then several data-mining constraints must be contented with expressive insights. Nevertheless, the survey reveals about distinct features of assimilated systems; thus, an informal analysis is made in health-based applications. Additionally, a complete library

package is essential for the proper movement of biomedical signals where the sequence of relationships is developed in this stage.

A proper data-mining methodology was listed in Reference [6], where subsets are utilized in a proper way, using fuzzy interference system. This provides a high advantage for the biomedical signal processing techniques, as a clustering procedure is applied with the best accurate results. Moreover, all of the data are dependent on the best combination rate, using comprehensive testing models. Since the clustering process is used, there is a high possibility of uncertainty in the detection of biomedical signals without a central procedure. To reduce the amount of uncertainty, fuzzy logic was applied in a data-analytics process as descriptive studies with four different types of data, namely text, image, signal, and experimental, integrated with central procedures [7]. Due to the incorporation procedure, large quantities are damaged, and they are converted to smaller quantities. However, the breaking procedure has several constraints, and non-fuzzy values are created in this process. Consequently, instead of using central procedures, multiple angle detection can be made with high security, wherein the data can be prevented to large extent [8]. As multiple angles are involved, a recapitulate table form is needed to handle diverse technology characteristics. Instead of using a separate table, the arrangement will appear better if a graph theory is implemented [9]. In the case of graphical representations, three types of analytics, namely expressive, prognostic, and authoritarian, can be defined with a decision-making approach. However, then again, the major drawback on analytic procedures without storage technique exists at this point, and it can be avoided only by providing a secondary procedure.

In advance, a type-two fuzzy model was created for medical informatics to handle both parallel and distributed information, using mobile agents [10]. The use of mobile agents separates the primary components, thus giving intensification to sub-processing technologies in the system. Moreover, with sub-processing blocks, a cognitive technology was specified with considerable requirement procedures. Meanwhile, the use of cognitive technology provides high nurture on mathematical representations by using binary fuzzy logic [11]. This type of binary representations can be handled by using stochastic and deterministic representations at a complete interval period of 0 and 1. To extend the binary boundary conditions, a knowledge representation of a medical data set was created with a classifying mechanism [12] where effective detection of severity problems is delivered. A dependable computing tool is implemented in the big data analytics [13], where the model is inexpensive with respect to the detection of different disease types. This knowledge representation is a six-stage process, with transformation applied at the middle stage, using selection procedures.

Instead of using staging procedures, an associated learning model can be developed for extending the operation with fuzzy set operations [14]. This, in turn, provides solutions to long-term problems where many classifiers can be recognized within a training process. By applying the long-term procedures, the proposed method determines the best accurate solutions, and score classification mechanisms are avoided. As a modification strategy, and to avoid drawbacks in existing methods, in the proposed method, a new system model is created that can be integrated with the fuzzy rule-based technique, using binary classifiers.

Recent Works from the Literature

Some of the researchers have also examined the effect of biomedical signals by using medical data and applying a Markov chain process in the integration process [15]. This type of Markov chain process provides effective results on tuning parameters, but when it is integrated with algorithmic cases or with a fuzzy interference mechanism, only low effectiveness can be achieved. Further developments are also made in medical paths for the detection of public health by using an e-learning technique, and it represents a knowledge representation path in a particular system [16]. However, the major drawback is that information that is related to the impact of COVID is discussed only with respect to learning techniques; thus, signal representations are not made. Therefore, to represent the

biomedical signals in the correct mode, the X-ray images are considered to be input [17] for the building up of the Hadoop infrastructure with a convolutional network. Still, the process of image processing by using a direct simulation setup is not considered, and this is observed as major drawback in the system. Furthermore, a medical analysis is carried out for transmitting the signals and identifying healthy persons in the absence of insomnia [18]. Nevertheless, the signals are separated, not processed, to a farther distance, and lossy signal representations are observed.

### 1.2. Research Gap and Motivation

All the analyzed works from the literature [1–18] used biomedical signals for medical applications with regard to different points of reference. Even many devices are built for data-gathering methodologies in existing models, and some drawbacks are also observed. However, in real-time implementation, most of the methods fail to analyze the effectiveness of biomedical signals by using the knowledge representation process. In the absence of fuzzy system models, the biomedical signals will have high interruption on the receiver side, and no other prevention techniques can be applied inside it. Moreover, one or a few methods have implemented the use of the Hadoop infrastructure for passing biomedical signals with high security and distance separation. In order to build the gap that is present in existing methods, the proposed method is implemented with a knowledge representation process.

In the proposed method, the applications of biomedical signals are processed by using an AI algorithm which indicates that automatic operation is carried out. To be more specific, a fuzzy interpretation mechanism is applied at the correlation state, using an activation function, and the output is indicated by using 0 s and 1 s. Furthermore, the knowledge representation of biomedical signals is carried out by using a Hadoop infrastructure wherein the loss of functionality of the signals is minimized. Moreover, a four-layer data model with fuzzy is incorporated for collecting the data at several authenticated health centers. The abovementioned process is usually carried out by using a real-time setup, and a working model for five different scenarios is also observed.

### 1.3. Objectives

The major contribution of the proposed model is to incorporate a fuzzy-based intelligent device with four-layer architecture in the presence of a Hadoop infrastructure to process the biomedical signals that satisfy the multi-objective functionalities as follows:

- To maximize the amount of signal strength that is used for recording the signals at various states, using a Hadoop structure.
- Preventing the loss functionality of signals that are transferred at larger distance with high separations inside the Hadoop infrastructure.
- To achieve a better correlation between incorporated fuzzy systems to make the signal travel at large-scale conditions at a low cost.

### 1.4. Paper Organization

The remaining sections of the paper are organized as follows. Section 2 presents a new system model for designing Hadoop infrastructure and for processing biomedical signal representations. Section 3 integrates the fuzzy system model with designed systems. Section 4 represents the outcomes of the integrated process under five different scenarios. Finally, Section 5 concludes the paper.

## 2. System Model: Data Mining Approach

In all biomedical applications, the major need is to gather the data from various sources for comparison cases, and any redundant data must be removed from the system. Therefore, the proposed technique provides high importance to data-mining procedures, in

addition to fuzzy incorporation systems. In the first step, the initial strength of biomedical signals must be monitored, and it can be mathematically represented by using Equation (1).

$$s_i = \sum_{i=1}^{n} \left| \frac{\alpha_i}{\alpha_n^s} \right| \times 100 \tag{1}$$

where $\alpha_i$ denotes the stored reference vector, and $\alpha_n^s$ represents the observed strength vector values.

From Equation (1), we can see that both vector values that are distributed in different regions are calculated, and a test is made at the initial stage. If the biomedical signal strengths are higher, then there is no necessity for boosting the process. During the testing process, the strength of the biomedical signals is much higher without any attenuation. Therefore, a cross-point intersection of signals can be formulated by using Equation (2), as follows:

$$\partial_i = \sum_{i=1}^{n} \frac{|M_i \cap \mu_i|}{\rho_i} \times 100 \tag{2}$$

where $M_i$ denotes data mining of $i$th set paths, $\mu_i$ represents the cipher score values, and $\rho_i$ indicates the maximum threshold values.

From Equation (2), a set of intersection points of different biomedical signals is observed and is later separated into distinct forms. This leads to the calculation of distance between two signals; as in many signal-processing techniques, the closer the signal levels are, the higher the accuracy of the fuzzy incorporated systems will be. Therefore, after intersection points, the distance measurements of biomedical signals are calculated by using Equation (3).

$$d_i = \sum_{i=1}^{n} \frac{ME_i - MR_{in}}{\max(ME_i, MR_{in})} \tag{3}$$

where $ME_i$ denotes the mean distance of signals, and $MR_{in}$ represents the input reference representation of signals.

Equation (3) denotes the difference between the mean distances that will provide variation, and, at this point, maximization can be performed within quick representation points. This maximization of points provide conjoint correlation between sets of points, and it can be mathematically modeled by using Equation (4).

$$C_i = \sum_{i=1}^{n} \frac{\varepsilon_x \varepsilon_y - I_i}{\sqrt{\left(\varepsilon_x - \varepsilon_y\right)^2}} \tag{4}$$

where $\varepsilon_x$ and $\varepsilon_y$ denote the coefficient matrix of $x$ and $y$ parameters, respectively; and $I_i$ represents the independent correlation matrix.

From Equation (4), we can see that the square of difference between the coefficients will provide higher values; thus, the values in the upper segment will rise to a larger extent. Therefore, the criteria for selecting signals can be given as follows:

$$P_i = \sum_{i=1}^{n} \varepsilon_i^2 - \left[ \varepsilon_n^2, \varepsilon_{in}^2 \right] \tag{5}$$

where $\varepsilon_i$, $\varepsilon_n$, and $\varepsilon_{in}$ denote the dimensional feature set of input and output variables.

The process of selecting signals will vary for each iteration; hence, Equation (5) can be redesigned in mathematical terms, where the signals can be differentiated by using Equation (6), as follows:

$$\varepsilon_{in} = \sum_{i=1}^{n} \left| \frac{1}{\gamma} \right| (l_i) \tag{6}$$

where $\gamma$ indicates the absolute signaling factor terms, and $l_i$ denotes the length of total signal representations.

The membership set can be used for defining a set of biomedical signal representations where multiple inputs and outputs can be defined. This can be expressed in mathematical form, as follows:

$$DF_i = \sum_{i=1}^{n} \frac{M_i(O_i) * O_i}{O_i^2} \tag{7}$$

where $M_i(O_i)$ denotes the membership function in de-fuzzified output at the receiver, and $O_i$ represents the output functions.

All the equations that are represented in the system model are implanted with fuzzy logic under three different levels: high, medium, and low. Therefore, the integration process is discussed in Section 3.

## 3. Optimization Using Fuzzy Systems

The major advantage of using the Fuzzy Interface System Algorithm (FISA) is that the input properties of biomedical signals are not clear; thus, the system produces very low robustness. To increase the robustness of the proposed method, precise inputs are evaded, thus integrating only supporting inputs that avoid fault data in the system [19–21]. Since FISA has low imprecise data, the integrated sensor will function effectively, and if any fault is identified, the sensor function stops with immediate effect. In this case, the program can be changed with respect to specification tuning parameters, wherein the cost of such changes will be much lower. Thus, the membership of fuzzy can be represented by using upper and lower bounds as follows:

$$M_f = \sum_{i=1}^{n} \frac{ds_l + ds_h}{2} \tag{8}$$

where $ds_l$ and $ds_h$ denote the data set in lower and upper boundaries.

Using the boundaries in Equation (8), a minimization function can be directed by using a coefficient matrix term wherein the centroid method is implemented. This can be represented as a two-unit exemplification, using Equation (9):

$$coeff_i = \sum_{i=1}^{n} o_{ik} \sum_{x=1}^{z} dist_{xz} \tag{9}$$

where $o_{ik}$ denotes the coefficient factor of the $i$th and $k$th terms; and $dist_{xz}$ represents the distance between the $x$ and $z$ sub-points.

The above equation can be linked to a trajectory function by using cluster weights that are represented by using Equation (10):

$$\sum_{i=1}^{n} o_{ik} \sum_{x=1}^{z} dist_{xz} = 1 \; for \; i, n, x, z > 0 \tag{10}$$

Whenever fuzzy set relationships are established, the binary form of fuzzy must be defined by using a constant integer set as a simple solution that can be achieved at sequential time steps. This can be represented by using Equation (11):

$$fz_{i+1} = \sum_{i=1}^{n} (T_i * T_n) + act_{in}(i+1) \tag{11}$$

where $T_i$ and $T_n$ represent the threshold limits of minimum and maximum cases, respectively; and $act_{in}$ denotes the activation function.

In Equation (11), the activation function is expressed in terms of the next periodic function, as the presence of uncontrolled values can be represented by using a stringent function, which is exponential in nature. These exponential characteristics can be defined by using Equation (12):

$$act_{in}(r) = \sum_{i=1}^{n} \frac{1}{(1 + e^{-r_{in}})} \tag{12}$$

where $r_{in}$ denotes the initial input activation constant, which is assumed to be of 15 periods.

In Equation (12), the activation function will remain active for a period of 15 s, and, after a certain time, it will automatically reach the permeation state, as power consumption during signal transmission can be reduced to larger extent. Therefore, to reduce the loss in signal, a square function must be defined in logic functionalities, which can be expressed by using Equation (13):

$$loss_i = min \sum_{i=1}^{n} \frac{n_s(i)}{n_d(i)} * \theta_i^2 \tag{13}$$

where $\theta_i^2$ denotes the square function; and $n_s$ and $n_d$ represents number of signals and database signals, respectively.

If Equation (13) is minimized, then the accuracy of proposed method can be increased to a great extent. However, it cannot be implemented in the same constraint as a single case; therefore, it is integrated with loop Equations, as shown in Figure 2. The combined effective outcomes are discussed in the subsequent section.

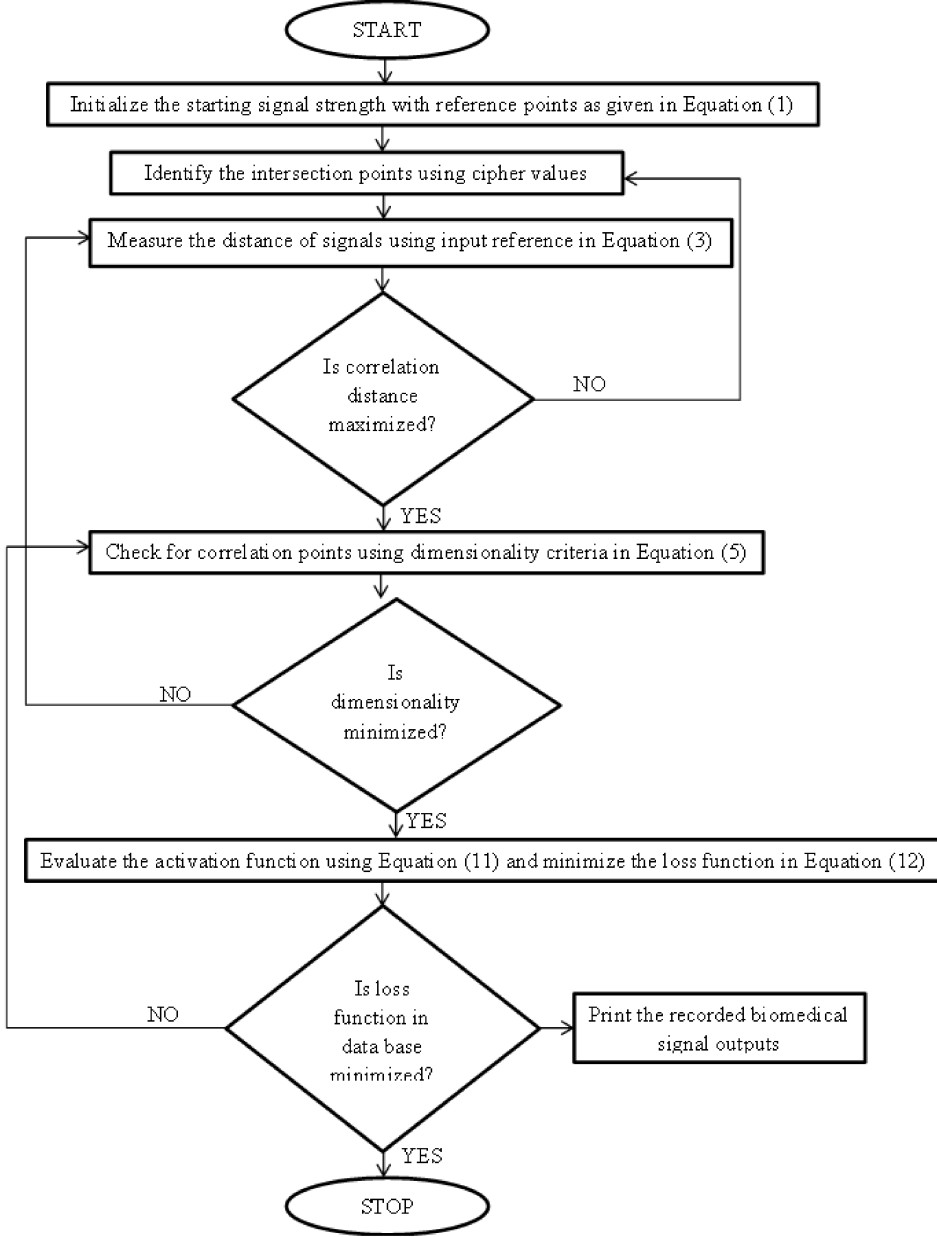

**Figure 2.** Fuzzy intelligence system for biomedical signal representations.

## 4. Results and Discussion

The output values of the developed system model are integrated for the purpose of data analysis using fuzzy systems. Since, in fuzzy systems, binary representations are made, probabilistic determinations will be made for decisions, thus dividing the input ecological data at maximized signal distance. In order to establish a complete relationship with knowledge representation, it is necessary to implement the data that are processed with master and slave blocks in a Hadoop infrastructure, using fuzzy interface systems. Thus, the developed system model is integrated with loop mechanisms with a high correlation matrix, using activation functions. Furthermore, the integrated process will have high similarities for all divided clusters, but this is eliminated by using fuzzy interface systems. This makes the signals to travel at maximized distance without any duplication representations. The outcome of the integrated system model a with fuzzy system is deliberated with simulation analysis in this section, where a Hadoop physical system is represented with a cluster formation technique. Moreover, for the tranquil analysis, the input and output nodes are separated, and individual node configurations are added in the specifications, where, in both nodes, the biomedical signals are passed at various points. During this signal flow point, three stages are designed, namely incorporation, dispensation, and cognitive therapy. For all of the abovementioned stages, a set of layered tools are organized, and they are examined under five different scenarios as follows:

Scenario 1: Amount of signal strength;
Scenario 2: Measurement of signal distance;
Scenario 3: Minimization of loss function;
Scenario 4: Fuzzy-signal correlation;
Scenario 5: Cost of implementation.

All five scenarios will exhibit unique characteristics in the designed system model; therefore, a high-end processor is used for simulation with MATLAB. Furthermore, for biomedical signal transmission, reception data are collected from the control center; therefore, more readings are congregated to examine the possible outcomes. The configuration setup with a data set is given in Table 1.

**Table 1.** Data set inceptions.

| Number of Patients | Reference Data Set | | | |
|---|---|---|---|---|
| | Total Volume of the Signal | Percentage of Signal Error | Percentage of Signal Projection | Position of Data Signals |
| 10 | 222.23 | 0.3 | +1.78 | −0.56 |
| 100 | 2356.15 | 1.4 | +1.94 | −0.28 |
| 1000 | 5891.01 | 1.6 | +2.15 | +0.01 |
| 2000 | 8942.07 | 2.0 | +2.28 | +0.07 |
| 5000 | 11,247.56 | 2.2 | +3.16 | +1.21 |
| 10,000 | 14,892.18 | 2.4 | +3.49 | +1.45 |
| **Signal characteristics** | | | | |
| Impetus of signal: 50 Volts | | | | |
| Time series: 200–400 microseconds | | | | |
| Input velocity: 20 m/s | | | | |
| Bandwidth: 20,000 Hz | | | | |
| Length of the biological signal (input): 25 micro volts | | | | |
| Length of the biological signal (output): 18 micro volts | | | | |
| Number of signals represented: 1768 | | | | |
| Mean period: 6001 s | | | | |

The biomedical signals that are represented in this process consist of time-series representations of all medical events in biological arrangement. By using such arrangements, it is possible to record all activities inside the body, such as the temperament rate, muscle contradictions, etc. The abovementioned monitored signals are obtained by analyzing the materials that are present inside the body systems, using a biomedical signal scanner.

- Scenario 1

Since the data that are collected from the patients are numerical in character, they cannot be processed at various stages, due to incomplete information in the central database. As a result, the data-mining methodology cannot be applied at various stages, as only manual changing process is present. To avoid the aforementioned condition and to convert the representation of the system, a signal representation technique is followed in medical treatment. This automatic procedure provides complete details of persistent characteristics where a sensor is used in the intermediate process as a channel to sense the values, and it reports the same to the control center. Moreover, the sensor can sense the received signals only when the signal strength is much higher; otherwise, intermediate noise will be present that will occupy the entire space in the system. The automatic signal representation is very important, as it gives rise to the perception learning mechanism, using fuzzy set models with limited exploration. Furthermore, if perception learning is precise with the maximization of signal strength, then the proposed system will provide a clear insight about classification features.

Figure 3 and Table 2 show the signal strength of the acquired data that can be visualized by using normalized frequency values. Before plotting the strength of the signal, the noise present in the system is removed and filtered by using the fuzzy least-square method. Therefore, the system is completely set as noise-free, and the received signal strength is maximized. This can be observed from the number of signal records, as it increases to a high amount; the low signal representations are transformed to high signals in both existing [6] and proposed models. However, the augmentation of signal strength in the proposed fuzzy model is much higher, as the Hadoop physical structure is incorporated in the system. Furthermore, if the signal does not reach a maximum value, it is stationary at average values, thus maintaining complete operation at a minimized risk factor.

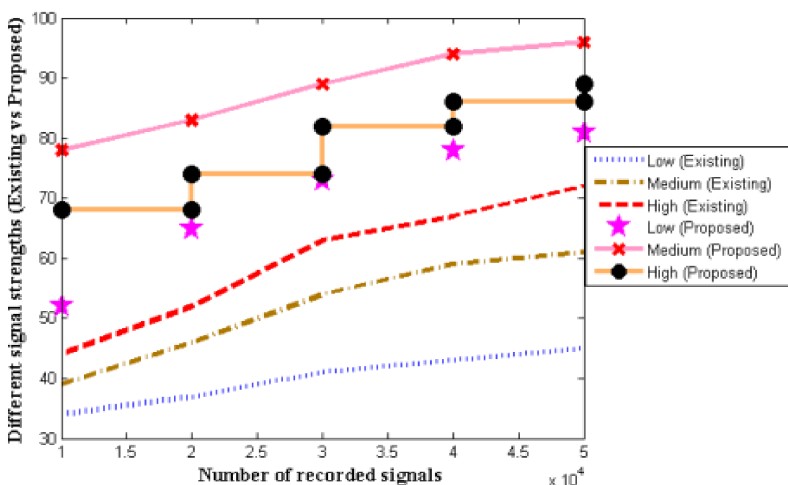

**Figure 3.** Measurement and comparison of signal strength.

Since the process carried out in the proposed method is discussed with the received signal, it is classified with low, center (medium), and upper strengths. All three values specify that, when the process is integrated by using the fuzzy set, it should not be lesser or greater than the reference data set of signals that is indicated in Table 1. The major reason for dividing these values is that a greater number of signals will be passed if the number of

patients is higher; therefore, to avoid correlations with different signals, they are grouped as low-, medium-, and high-range signals.

**Table 2.** Variation of signal strength.

| Number of Signal Records | Low [6] | Medium [6] | High [6] | Low | Medium | High |
|---|---|---|---|---|---|---|
| 10,000 | 34 | 39 | 44 | 52 | 68 | 78 |
| 20,000 | 37 | 46 | 52 | 65 | 74 | 83 |
| 30,000 | 41 | 54 | 63 | 73 | 82 | 89 |
| 40,000 | 43 | 59 | 67 | 78 | 86 | 94 |
| 50,000 | 45 | 61 | 72 | 81 | 89 | 96 |

- Scenario 2

Due to incorporation of fuzzy set an intersection occurs in signal transmission process and these intersection points must be different for all signals. Therefore, a distance transmission technique is incorporated for biomedical signals for recording multiple values in the system. The distance of biomedical signals is measured by using the installed electrodes at several points in real-time implementation. For each time period the electrodes will change the position as the signal type will differ at applied input side. Moreover, fuzzy systems will develop a set of distance with respect to signals which are measured using a serrate electrode. Both the values of electrodes are combined as mean values as given in Equation (3). Furthermore, the distance of measurement is separated by using sample frequency values for processing the signals, which are represented as reference representations. The installed electrodes also vary according to the number of signals that are received from different patients. The entire system was notified once the values are exceeded with compared reference values. In such a case, the system can be prevented from entering in the same node; therefore, signal distance can be measured at this point. If the signals are closer at the intersection point, then a high risk factor will be assured, whereas, if the signals are farther away from intersection points, the risk factor can be reduced to a great extent. Usually, the signals are measured from bonce position to the bottom of the pole by using a data-acquisition device; therefore, it is better to incorporate the points at the first stage, before measurement. This gives a clear insight about distance, and if any wrong values are measured, then points can be changed with immediate effect.

Figure 4 and Table 3 show the simulated distance of biomedical signals with two pole points, and in the proposed method, the points are not varied until maximum signal transmission is observed. The varying distance is measured for time periods that start at 6:00 h and end at 24:00 h, and in this intermediate period, the distance of measurement cannot be varied, as the tolerant limits remain in a steady condition. Moreover, the separated clusters will have individual separated distance; thus, they are grouped within the primary cluster. It can be observed that, even during changing hours, the distance of separation is maximized for the proposed method, whereas, in the existing system [6,15], the separated distance of all combined signals is much lower. Due to the low signal representations, a correlation factor is determined in the existing models, but low correlation factors exist in the fuzzy interference system. This proves that, at varying time factors, the proposed method can provide accurate signals at distant points without any interruption from other clustered signals.

**Table 3.** Distance of separation vs. time periods.

| Time Period | Distance Separation [6] | Distance Separation [15] | Distance Separation (Proposed) |
|---|---|---|---|
| 6 | 12.7 | 17.8 | 36.8 |
| 7 | 12.4 | 17.9 | 34.9 |
| 8 | 12.1 | 16.4 | 32.1 |
| 9 | 11.7 | 16.9 | 30.4 |

**Table 3.** *Cont.*

| Time Period | Distance Separation [6] | Distance Separation [15] | Distance Separation (Proposed) |
|---|---|---|---|
| 10 | 11.2 | 14.1 | 29.3 |
| 11 | 10.4 | 15.6 | 28.7 |
| 12 | 10.1 | 17.2 | 27 |
| 13 | 9.6 | 15.3 | 26.5 |
| 14 | 8.3 | 13.1 | 25.2 |
| 15 | 7.4 | 12.5 | 24.6 |
| 16 | 6.9 | 11.9 | 23.2 |
| 17 | 5.1 | 10.1 | 22.7 |
| 18 | 4.4 | 9.3 | 21.8 |
| 19 | 4 | 8.4 | 20.6 |
| 20 | 3.8 | 8.1 | 19.2 |
| 21 | 3.2 | 7.8 | 18 |
| 22 | 2.6 | 7.5 | 17.5 |
| 23 | 2.2 | 6.7 | 16.3 |
| 24 | 1.26 | 6.1 | 15.9 |

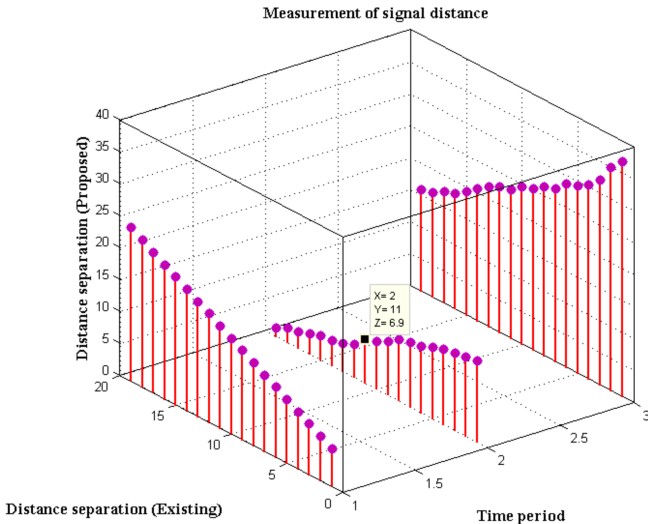

**Figure 4.** Comparison of signal separation with distance measurements.

- Scenario 3

The handling capability of biomedical signals using fuzzy system can be proved by using a loss-function parametric evaluation, wherein low loss must be achieved. This loss function can be determined by using the database reference set and square of logical functionalities, wherein the enhanced signal outputs are represented. During this determination process, the loss function is mapped with real period representations that are associated with various clusters. However, a least-square technique can be applied in this case to prevent such signal losses in the system. In addition, the distance of measurement that is observed in Scenario 2 is also higher; therefore, the proposed system will provide a much lower loss, and the problem of uncertainty can also be solved. The biomedical signals exhibit two different properties, namely variance and symmetric, for which values in terms of magnitude are measured. Thus, the two property limits are examined and after careful experimentation error in loss values are found and plotted in Figure 5.

From Figure 5 and Table 4, it can be perceived that, if the errors are present at one side of the plot, it will be propagated through all distinct directions. The simulation plot is represented with a number of iterations, and for each variation, the total loss in the system is calculated. The major reason for plotting the number of iterations is that the expected output will be compared with achieved fuzzy set outputs, and if any variations

are observed, then the error for that particular period will be indicated. Furthermore, at high state iterations, the proposed method achieved zero loss conditions, as compared to the existing method [6,15], which is plotted without comparison, as data from other systems cannot be measured. In addition, the loss function can also be measured by using prediction values, but negative values in the biomedical systems are negligible if it falls below the value of 1. The loss values that are mentioned in Table 4 were calculated by using an analytical equation that was designed by using Equation (12). At the first stage, the database signals that are given in Table 1 and number of original signals are compared, thereafter providing the angular square function of transmitting signals.

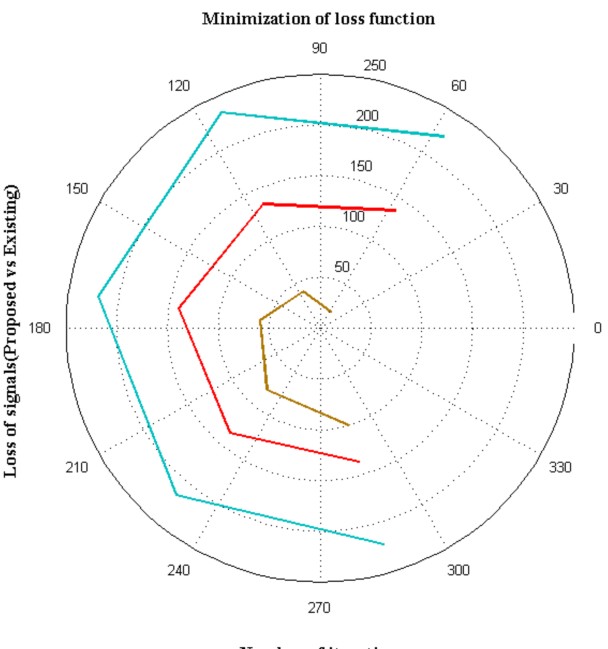

**Figure 5.** Association of loss function at minimized process.

**Table 4.** Comparison of loss rates.

| Number of Iterations | Loss [6] | Loss [15] | Loss (Proposed) |
|---|---|---|---|
| 20 | 225 | 179 | 138 |
| 40 | 234 | 171 | 135 |
| 60 | 221 | 168 | 141 |
| 80 | 217 | 163 | 136 |
| 100 | 222 | 158 | 137 |

- Scenario 4

In this scenario, the correlation of fuzzy signals is detected to represent the individual signals with a set of coefficient matrices. This type of correlation is applied in biomedical signals at both the transmission and reception stage, where the delays of the signals are measured. If the delay of a signal at transmission is high, then the coefficient matrix will have high cross-correlation, whereas, if the delay is less, then the correlation of signals can also be reduced. If the receiver is considered, the delay should be much less, as the data-mining process is handled at the receiver section for the fuzzy interpolation set. To have a low correlation matrix, intermediate rules are defined in the data set by using different scaling factors and a low sampling rate. Furthermore, for defining the matrix type, two different constant terms are indicated with distances from sub-points where membership function can be established. Using the defined function, the link that exists with different signals can be easily modified, and correct linking periods can be measured. The corrected linking periods are plotted in Figure 6.

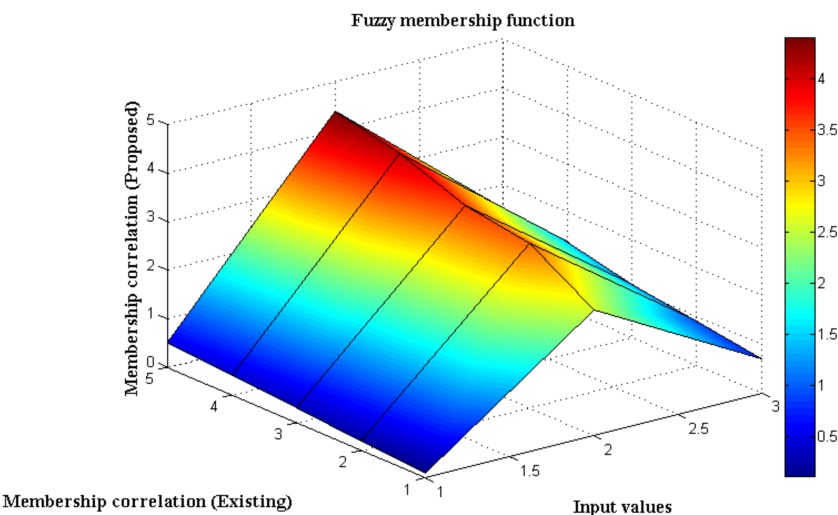

**Figure 6.** Rapport membership of fuzzy interference system.

From Figure 6 and Table 5, it can be detected that the correlation points of the fuzzy sets for biomedical signals provide good representation points for the projected system model, whereas an average set of points is obtained in existing models [6,15], as the intersection of membership functions are not defined. In the simulation point, the input values are changed from 0.1 to 0.5, and for each point, membership correlation is checked. During the initial period, the correlation is below the value of 1, whereas, after setting the input values to 0.3, the membership of the proposed fuzzy set crossed 1, and it is defined as an intermediate correlation set. However, for existing models, the correlation is much higher, and the signal rate, in this case, will be maximized, thus leading to a high state of power consumption. Conversely, at the final input stage, the proposed method provides only 1.5 values of parallel association between input and output links.

**Table 5.** Relationship of correlation: A comparison.

| Input Values | Membership Correlation [6] | Membership Correlation [15] | Membership Correlation (Proposed) |
|:---:|:---:|:---:|:---:|
| 0.1 | 2.6 | 2.1 | 0.7 |
| 0.2 | 3.4 | 2.4 | 0.9 |
| 0.3 | 3.6 | 2.8 | 1.1 |
| 0.4 | 4.1 | 3.2 | 1.4 |
| 0.5 | 4.4 | 3.6 | 1.5 |

- Scenario 5

In this scenario, the cost of the data-management system is calculated for managing a large volume of data. In Hadoop systems, the size of the physical infrastructure is much higher as compared to other systems; therefore, the cost of variation directly depends on the system. The basic cost of both the hardware and software specifications in the Hadoop system varies from 1000 dollars, whereas it increases for every technological standpoint. However, in the proposed system, the biomedical signals are transmitted inside the designed Hadoop infrastructure, and the clusters of Hadoop will transfer the connection to external pathways. Therefore, the total cost of installation is calculated with a number of clusters, sensors for signal transmission, and reception with Hadoop infrastructures. Furthermore, there is a need to store the data signals, as the information is necessary for the future; hence, the cost of a single disk management system is also added as an integration cost. The total cost of the designed system is simulated in Figure 7.

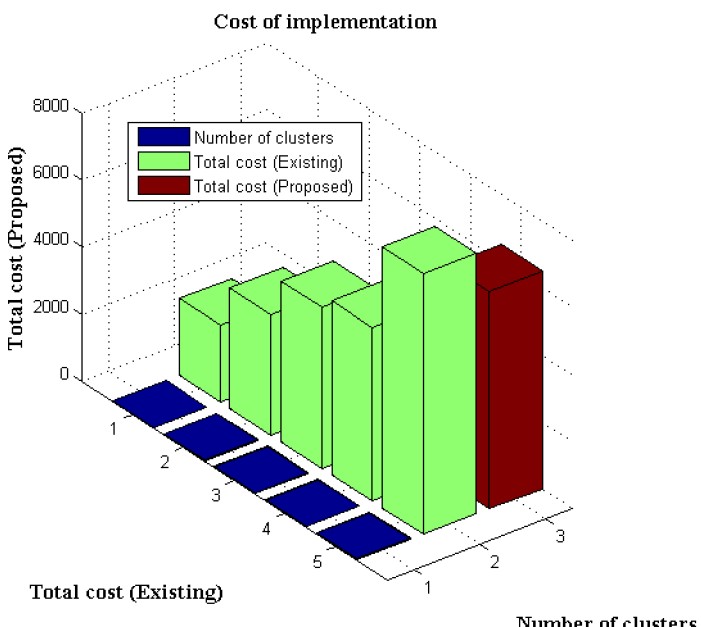

**Figure 7.** Imperial cost of comparison with cluster nodes.

From Figure 7 and Table 6, it can be observed that the setup cost is plotted for both the existing [6,15] and proposed method. In this comparison, the case number of clusters is varied, thus increasing the cost of implementation from initial to last state. Consequently, the proposed method implements much lower cost as compared to the existing method, for which, in the initial demonstration state, the process building cost only 1000 dollars. Then, updating with a greater number of clusters added extra cost with beneficial systems to the public, varying up to 6500 dollars. Under the same circumstances, the existing method adds a total cost of 7800 dollars, which is much higher due to the surplus infrastructure in the integration process.

**Table 6.** Cost of implementation.

| Number of Clusters | Total Cost [6] | Total Cost [15] | Total Cost (Proposed) |
| --- | --- | --- | --- |
| 6 | 2300 | 1890 | 1000 |
| 12 | 3600 | 2346 | 1240 |
| 18 | 4800 | 3279 | 2560 |
| 24 | 5200 | 4300 | 4200 |
| 30 | 7800 | 6900 | 6500 |

*Performance Analysis of FISA*

In addition to all the fundamental scenarios where both Hadoop and FISA were tested to prove their complete effectiveness, some added performance determines the complete characteristics of the proposed methods. Therefore, two case studies were reported in this section, using big-O-notation, which represents the complexity of time and space representations. Consequently, from defined objectives, the performance analysis must reduce the complexities that are present in a particular system; otherwise, the system throughput will be reduced, and the same system model cannot be implemented in real-time conditions. The complexities of FISA are simulated by using MATLAB by considering only the best iteration periods as clusters that are represented in the projected model.

- Space Complexity

The total space occupied by FISA is measured by using a sorting score method wherein secondary memory space is also built into the internal systems. In the case of the memory of FISA being much higher, if more parameters are updated with high iteration periods,

then the storage capacity will be protracted. Therefore, if primary memory is occupied, then data will be immediately stored in the secondary memory space, which adds additional advantage for FISA-implemented systems. Since the FISA is defined in another system model, the space complexity will be represented only by using data representation segments. In the proposed method, the variables that are represented in data representation will be stored by using 8 bytes, and the return value will be definite by using array of elements. Thus, the space complexity is simulated and represented in Figure 8 and Table 7.

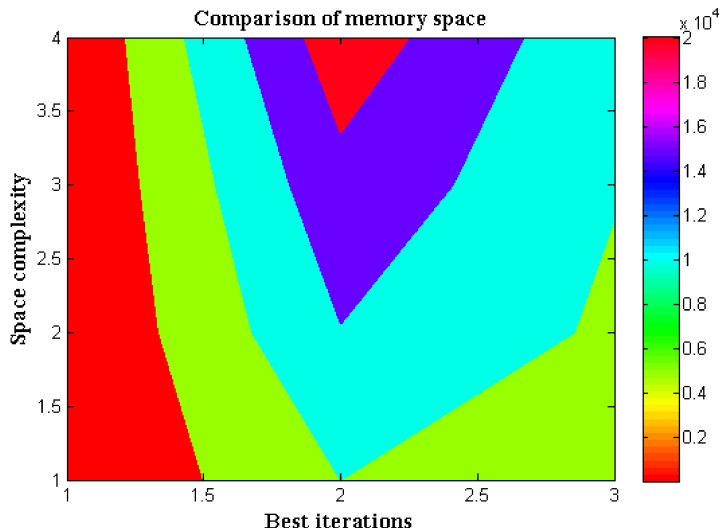

**Figure 8.** Comparison of memory requirements: FISA vs. traditional algorithms.

**Table 7.** Comparison of storage capacity.

| Best Iterations | Space Complexity [9] | Space Complexity (Proposed) |
| --- | --- | --- |
| 20 | 10,000 | 8500 |
| 50 | 14,800 | 9200 |
| 80 | 18,380 | 10,230 |
| 100 | 23,000 | 11,000 |

- Time Complexity

Since the proposed method is implemented with a greater number of operations, it is necessary to monitor the number of operations in a particular period. This is measured by using time complexity, where only a fixed amount of time is allocated to a particular operation for performing its complete action. In addition, all created loops with a FISA system model are running in a parallel stream; therefore, it is essential to avoid complexities in time periods. The time complexity of FISA is implemented by using conditional statements where the time periods are linear in nature. Furthermore, if any worst-case periods are found, then high importance will be given to convert it into either medium- of high-speed conditions, and if it cannot be converted, then those parameters will be neglected from the system. Furthermore, for the conversion process, the proposed system divides the data set into two different sets, where one set will be executed at the first half cycle and another set at another half cycle period. The time complexity values that are shown in Table 8 are simulated and deliberated in Figure 9.

**Table 8.** Comparison of time complexities.

| Number of Hadoop Infrastructure | Time Complexity [9] | Time Complexity (Proposed) |
|---|---|---|
| 100 | 65 | 30 |
| 200 | 68 | 28 |
| 300 | 62 | 24 |
| 400 | 69 | 22 |
| 500 | 61 | 20 |

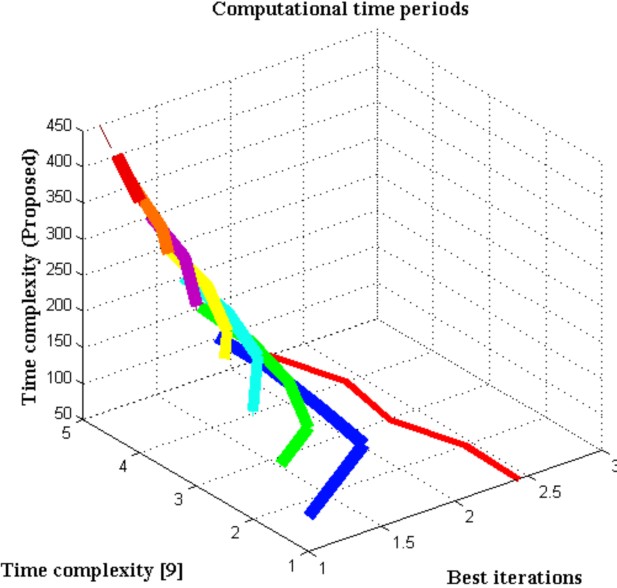

**Figure 9.** Comparison of time periods: FISA vs. traditional algorithms.

## 5. Conclusions

The current real-time issues that are faced by all healthcare management systems were addressed in this article. Till now, all the clinicians have been investing only sensor-based technologies, where, in turn, wireless modules can be created, and it can monitor the current parametric values of different patients. However, this monitoring does not provide any solutions, and, instead, it can operate on communication technology without any physical infrastructure. Therefore, a new attempt was taken for constructing a physical infrastructure by using Hadoop systems for transferring biomedical signals within internal clusters. The internal clusters will be divided into several clusters, and, as a result, the signals will be transferred to outdoor environments. The abovementioned process is carried out by using data mining methodologies wherein classification is processed by using several input images. Moreover, a correlation model is introduced into the system to avoid similarity index of input that is given at one end of the device. To process the Hadoop systems, a FISA algorithm is introduced in which time complexity is reduced to a much greater extent. In addition to prove the effectiveness of the proposed model, a five-scenario case study was examined, and the performance of FISA was compared with existing models. Since the amount of measured signal strength is higher, the proposed method can provide effective outcomes, while minimizing the of loss functions. In the future, the FISA integrated system model can be used without a clustering process, and a mobile application can be developed for input and output classifications.

**Author Contributions:** Data curation, T.H. and R.A.; writing original draft, H.M.; supervision, S.S. and M.U.; project administration, S.S.; conceptualization, H.M. and S.S.; Methodology, S.S. and H.M.; Validation, T.H. and R.A.; visualization, H.M. and S.S.; resources, T.H. and R.A.; review and editing, S.S. and M.U.; funding acquisition, M.S. and A.A. All authors have read and agreed to the published version of the manuscript.

**Funding:** We deeply acknowledge Taif University for supporting this study through Taif University Researchers Supporting Project number (TURSP-2020/115), Taif University, Taif, Saudi Arabia.

**Institutional Review Board Statement:** Not applicable.

**Informed Consent Statement:** Not applicable.

**Data Availability Statement:** Not applicable.

**Conflicts of Interest:** The authors declare no conflict of interest.

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
