# Peer review of "Biomedical Signals for Healthcare Using Hadoop Infrastructure with Artificial Intelligence and Fuzzy Logic Interpretation"

_applsci, doi:10.3390/app12105097_

Round 1

Reviewer 1 Report

This article addresses the issues of physical infrastructure using hadoop based systems where a four layer model is created. Concretely, a new attempt has been taken for constructing a physical infrastructure using hadoop systems for transferring bio medical signals within internal clusters. Generally, the proposal is of some novelty and significance.

The following comments should be considered to improve the quality.

  1. The major contributions of the paper should be highlighted explicitly in the 1st section.
  2. Paper organization structure should be added at the end of the 1st section.
  3. The authors should further adjust the format according to the journal style, especially in terms of punctuation.
  4. Figure 1 is a bit vague. A clearer figure is needed to replace it.
  5. In the experiment evaluation section, more details should be included such as experiment configurations, parameter settings, experiment repeation times and so on.
  6. Proofread the whole paper and correct the existing spelling errors in terms of grammar and syntax.
  7. Medical data analysis studied in this paper has also been investigated by many other researchers. Therefore, the authors should also introduce the following related literatures: A Chan-Vese Model Based on the Markov Chain for Unsupervised Medical Image Segmentation; Effect of E-Learning on Public Health and Environment During COVID-19 Lockdown; Heart-Rate Analysis of Healthy and Insomnia Groups with Detrended Fractal Dimension Feature in Edge; Diagnosis of COVID-19 from Chest X-Ray Images Using Wavelets-Based Depthwise Convolution Network.

Author Response

PFA.

Reviewer 2 Report

Authors study Bio-Medical Signals for Health Care and try to address the issues of physical infrastructure using Hadoop based systems. I have some comments and questions that considering them might improve the quality of this research. 

1- The readability of this paper needs improvement. The importance of this research, methodology, the contribution of the authors, and many essential aspects are missing. The paper needs a deep review and re-writing, especially in the Introduction. 

2- Authors start the paper with a literature review instead of a short introduction in which they can explain the organization of the paper, their presented novelties, and also the gained results compared with earlier methods.

3- The presented methods in the literature review need to be addressed in a systematic manner. It is expected that interested readers can follow the history of the research and understand the existing issues that this paper is trying to resolve. It is expected that some of the existing most-related recent methods are presented in deep so that the comparison is fair. 

4- As indicated in previous comments, the novelty of this research is missing. Authors need to add a separate paragraph at the end of the Introduction section and clearly present their contributions (novelties) to the literature. 

5- The quality of all figures needs improvement (current figures are unacceptable in terms of quality). 

6- Authors state that "The outcome of the integrated system model with fuzzy system...". Therefore the reader is expecting a connected and related explanation in which the existing system model and the contribution of the authors are clearly presented. The system model and fuzzy optimization need clarification. 

7- Authors emphasize biomedical signals in this paper. The used dataset needs to be clearly presented for different scenarios. It is recommended to use existing datasets or publish the used dataset for public access and improvement of this research. 

8- In figure 2, the authors need to draw the existing method and proposed results (assuming that they have clearly explained before, what is the proposed method), in one figure to make them comparable. Currently, they are illustrated in two separate sub-plots. 

9- Authors need to clearly explain the meaning of Low, medium, and high both in a mathematical and general manner. 

10- Authors need to use more proper captions for the figures so that they could be interpreted without referring to the text. For instance, see figure 3, "distance of signal separation".

11- In figure 3 and table 2, how do authors simulate the distance of biomedical signals?

12- In table 3, the authors use the word proposed for loss. Authors need to clearly relate the proposed equation number for this kind of statement.

13- As can be seen, the authors compare their results only with [6]. This reference is published in 2008!. Authors need to enrich the literature review (according to previous comments) and select some of the most recent existing methods for comparison. 

14- There are many comments remaining, however, most of them are similar to the ones already stated. It is expected that the authors fully revise the paper (not only the mentioned comments) if they get the chance to send a revision. 

Author Response

PFA.

Round 2

Reviewer 2 Report

Authors have tried to answer the comments and questions. 

1- There are still some Typos and grammatical errors. For instance line 167, "signals is...". Authors need to carefully read the paper for all typos and grammatical errors. Or also in the title of Table 1. 

2- Authors need to clearly state that what are these biomed signals, how they are obtained and etc. Authors also need to specify the signal characteristics including length, packet description and related titles. 

3- Authors need to specify the the dimension for titles in Table 1. For instance "Total volume of the signal" is based on GB or MB or Number of signals or ....?

Author Response

PFA.